# Oleanolic Acid Promotes the Formation of Probiotic *Escherichia coli* Nissle 1917 (EcN) Biofilm by Inhibiting Bacterial Motility

**DOI:** 10.3390/microorganisms12061097

**Published:** 2024-05-29

**Authors:** Dan Liu, Jingjing Liu, Lei Ran, Zhuo Yang, Yuzhang He, Hongzao Yang, Yuandi Yu, Lizhi Fu, Maixun Zhu, Hongwei Chen

**Affiliations:** 1College of Veterinary Medicine, Southwest University, Chongqing 402460, China; liuliu64521@126.com (D.L.); jingjing0308@126.com (J.L.); 15086766496@163.com (L.R.); yangzhuo202304@163.com (Z.Y.); hyzlucky250@gmail.com (Y.H.); yhz03008@swu.edu.cn (H.Y.); 2National Center of Technology Innovation for Pigs, Chongqing 402460, China; yuyd@cqaa.cn (Y.Y.); fulz@cqaa.cn (L.F.); zhumx@cqaa.cn (M.Z.); 3Chongqing Academy of Animal Sciences, Chongqing 402460, China; 4Traditional Chinese Veterinary Research Institute, Southwest University, Chongqing 402460, China

**Keywords:** probiotics, *Escherichia coli* Nissle 1917, oleanolic acid, biofilm, adhesion, iron uptake

## Abstract

Probiotic biofilms have been beneficial in the fight against infections, restoring the equilibrium of the host’s gut microbiota, and enhancing host health. They are considered a novel strategy for probiotic gut colonization. In this case, we evaluated the effects of various active substances from traditional Chinese medicine on *Escherichia coli* Nissle 1917 (EcN) to determine if they promote biofilm formation. It was shown that 8–64 μg/mL of oleanolic acid increased the development of EcN biofilm. Additionally, we observed that oleanolic acid can effectively suppress biofilm formation in pathogenic bacteria such as *Salmonella* and *Staphylococcus aureus.* Next, we assessed the amount of EcN extracellular polysaccharides, the number of live bacteria, their metabolic activity, the hydrophobicity of their surface, and the shape of their biofilms using laser confocal microscopy. Through transcriptome analysis, a total of 349 differentially expressed genes were identified, comprising 134 upregulated and 215 downregulated genes. GO functional enrichment analysis and KEGG pathway enrichment analysis revealed that oleanolic acid functions are through the regulation of bacterial motility, the iron absorption system, the two-component system, and adhesion pathways. These findings suggest that the main effects of oleanolic acid are to prevent bacterial motility, increase initial adhesion, and encourage the development of EcN biofilms. In addition, oleanolic acid interacts with iron absorption to cooperatively control the production of EcN biofilms within an optimal concentration range. Taking these results together, this study suggests that oleanolic acid may enhance probiotic biofilm formation in the intestines, presenting new avenues for probiotic product development.

## 1. Introduction

The World Health Organization defines probiotics as “living microorganisms that, when present in sufficient quantities, can have beneficial effects on the body’s health” [1]. Probiotics, as non-toxic, side-effect-free, and residue-free high-quality feed additives, typically come to the public’s attention as food supplements, nutritional supplements, and other health supplements. Probiotics have the function of preventing and treating diseases. They can enhance the immune system; prevent intestinal infections, diarrhea, irritable bowel syndrome, and necrotizing enterocolitis; and stabilize the intestinal mucosal barrier [2,3]. The majority of probiotics in the human body are primarily located in the intestine, and their ability to confer beneficial effects hinges on successful colonization within this region. A recent investigation involved the oral administration of *Clostridium butyricum* to mice for a duration of one week, which resulted in sustained colonization levels for the subsequent six days. This intervention led to an elevation in the Firmicutes to Bacteroidetes ratio within the gut microbiota, along with a notable increase in the levels of advantageous short-chain fatty acids in the colon [4]. Studies have suggested that following oral intake and discontinuation of probiotic consumption, a significant portion of these microorganisms are expelled from the colon through fecal matter [5]. The clinical effectiveness of probiotics is often hindered by their limited capacity to effectively establish colonization within the intestine. Despite the widespread practice of oral administration for probiotic consumption, their bioavailability is frequently constrained by the hostile conditions present in the gastrointestinal tract. Biofilm can be defined as a microbial community embedded in a self-produced extracellular polymeric matrix (EPS), attached to biological or abiotic surfaces. It is a complex, highly hydrated three-dimensional structure [6]. At present, research on biofilms mainly focuses on biofilm-related infections caused by pathogens, while research on probiotic biofilms is in its early stages.

*Escherichia coli* Nissle 1917 (EcN) is a Gram-negative probiotic strain that was originally discovered by the German physician Alfred Nissle from the fecal sample of a soldier who demonstrated resistance to intestinal bacterial infections caused by *Shigella* [7]. EcN is frequently utilized in the management of gastrointestinal disorders, such as diarrhea, inflammatory bowel disease, and ulcerative colitis [8]. Additionally, EcN exhibits immunomodulatory properties by enhancing the production of immunoglobulins and activating lymphocytes upon intestinal colonization. It triggers specific humoral and cellular immune responses and promotes non-specific innate immunity [9]. EcN is equipped with three distinct types of pili, namely F1A, F1C, and curli, which facilitate the attachment to intestinal epithelial cells and the development of biofilms. This mechanism establishes a protective microbial barrier on the intestinal mucosa, thereby preventing the infiltration of pathogenic microorganisms [10]. There are reports indicating that the formation of probiotic biofilms is crucial for the stable presence of the ecosystem in the body, particularly in suppressing the growth and adhesion of pathogens [11]. Biofilm probiotics have a stronger immune regulatory effect than probiotics in a planktonic state [12]. Compared with conventional probiotic preparations, probiotics in a dense biofilm state have stronger resistance to freeze-drying, heat, and acid [13]. Plant-derived active substances are the primary source of new drug development, ranking second only to other natural drug sources and synthetic chemicals [14].

Research has found that trans-resveratrol can enhance the adhesion ability of *Lactobacillus paracasei* ATCC334, thereby promoting its biofilm formation. While promoting bacterial aggregation, it does not cause pro-inflammatory reactions in intestinal epithelial cells [15]. At present, utilizing natural compounds from plant sources to boost the formation of probiotic biofilms has been demonstrated as a promising strategy for preventing and treating various biofilm-related infections [16]. Therefore, this study took probiotic EcN as the test bacterium and screened effective active ingredients from traditional Chinese medicine by evaluating this activity, that is, promoting the formation of probiotic EcN biofilms while inhibiting the formation of pathogenic biofilms. Through a lot of preliminary screening work, we found that oleanolic acid has obvious effects. The oleanolic acid studied in this article is a pentacyclic triterpenoid compound commonly found in the plant kingdom. It is widely used due to its potential anticancer, hepatoprotective, lipid-lowering, antioxidant, antibacterial, and other effects [17,18,19]. Oleanolic acid has demonstrated the ability to impede the development of biofilms produced by cariogenic bacteria, such as *Streptococcus mutans*, suggesting its potential utility as an antibacterial agent in combating dental caries [20]. Furthermore, oleanolic acid has been found to impact the virulence and biofilm formation of *Listeria monocytogenes* as well [21]. However, there is currently no report on the effect of oleanolic acid on the formation of EcN biofilms. In a preliminary systematic assessment, we examined the impacts of different active components found in traditional Chinese medicine on the EcN biofilm. Our results suggest that solely oleanolic acid has the ability to promote the development of EcN biofilm while concurrently impeding the formation of biofilm by pathogenic bacteria. Additionally, the study aims to investigate the mechanisms of action of oleanolic acid, providing a theoretical basis for the development of novel and efficient probiotic products for intestinal colonization.

## 2. Materials and Methods

### 2.1. Materials

Oleanolic acid (OA)with a purity of 97% was stored at 4 °C and was purchased from Shanghai McLean Biochemical Technology Co., Ltd. The OA reserve solution in this experiment was dissolved in dimethyl sulfoxide. *Escherichia coli* Nissle 1917 was purchased from the China General Microbiological Culture Collection Center. EcN was inoculated into MH medium and incubated on a shaking table for 5–6 h to reach the logarithmic growth phase, while *Salmonella* 011 from pigs and *Staphylococcus aureus* 002 from sheep were isolated and preserved in the College of Veterinary Medicine at Southwest University. The strains were stored at ^−8^0 °C in 60% glycerol broth for seed preservation. Bacterial strains were separated and cultivated by streaking them on agar plates into three zones, and stored at 4 °C. Each agar plate was used for about a week to maintain its bacterial vitality. The UV–visible spectrophotometer was purchased from Hanyi Instrument (Shanghai) Co., Ltd. (Shanghai, China).

### 2.2. Determination of EcN Biofilm Formation Curve

EcN in Mueller–Hinton (MH) medium was inoculated and allowed to reach the logarithmic growth phase at a concentration of 10^8^ colony-forming units per milliliter (CFU/mL). Subsequently, 100 μL of the bacterial suspension was dispensed into sterile 96-well plates. The plates were incubated at 37 °C for durations of 6, 18, 24, and 48 h to establish biofilm models. After biofilm formation, it was quantified using crystal violet staining. Then, 100 μL of PBS was washed three times to remove non-adhering planktonic bacteria. Next, 100 μL of methanol was fixed for 10 min, and the biofilm was stained with 0.04% crystal violet for 20 min. Finally, they were dissolved in 100 μL of an acetic acid solution. The OD value was measured at 600 nm.

### 2.3. Determination of Minimal Inhibitory Concentration

The minimum inhibitory concentration was determined based on the microdilution method described in [22]. MIC refers to the minimum concentration of drugs that can completely inhibit bacterial growth. In a 96-well plate, 40 μL of MH broth was dispensed into columns 1–12. Column 1 received 40 μL of oleanolic acid at a concentration of 8192 μg/mL, which was then thoroughly mixed by gentle vortexing and subsequently diluted in a sequential manner up to the 10th column. A volume of 40 μL of the EcN working bacterial solution was transferred from column 9 to column 1 in a sequential manner. The 10th column served as the drug control, column 11 contained an additional 40 μL of bacterial solution as the positive control, and column 12 was supplemented with 40 μL of MH broth as the blank control. The plate was then incubated at 37 °C for 16 h, and the MIC was determined as the lowest concentration of oleanolic acid, at which no bacterial growth and clear culture medium were observed after the 16 h incubation period.

### 2.4. Growth Curves

The EcN overnight culture was diluted to a concentration of 10^8^ CFU/mL. To achieve final concentrations of 128, 64, 32, and 16 μg/mL of OA, a 6400 μg/mL stock solution of OA was prepared using DMSO as the solvent and diluted by factors of 50, 100, 200, and 400 using MH culture media. After dilution, 2 mL of various concentrations of OA were mixed with 2 mL of the ECN working solution and thoroughly combined. Shaker oscillation cultivation was conducted at 37 °C, and the OD value was measured at 600 nm using a spectrophotometer at 0, 2, 4, 6, 8, 10, 12, and 24 h. Within a specific range of wavelengths, the absorbance value was directly proportional to the bacterial concentration. A growth curve was constructed by plotting the relationship between absorbance and incubation time. A culture medium with a 2% DMSO solution was used as the control.

### 2.5. Biofilm Formation

Using the method described by [23] with some modifications, total biofilm formation was measured in a 96-well plate. To achieve final concentrations of 128, 64, 32, and 16 μg/mL of OA, a 6400 μg/mL stock solution of OA was prepared using DMSO as the solvent and diluted by factors of 50, 100, 200, and 400 using MH culture media. After dilution, 50 μL of each OA concentration was combined with 50 μL of the ECN working solution and thoroughly mixed. The resulting mixture was then transferred into a fresh 96-well plate and incubated at 37 °C for a period of 24 h. After biofilm formation, it was quantified using crystal violet staining. Then, 100 μL of PBS was washed three times to remove non-adhering planktonic bacteria. Next, 100 μL of methanol was fixed for 10 min, and the biofilm was stained with 0.04% crystal violet for 20 min. Finally, they were dissolved in 100 μL of an acetic acid solution. The OD value was measured at 600 nm. A culture medium was used with a 2% DMSO solution as the control group. The method for determining biofilm formation of *Salmonella* 011 and *Staphylococcus aureus* 002 was the same as described above.

### 2.6. MTT Assay

3-(4,5-Dimethylthiazol-2-yl)-2,5-diphenyltetrazolium bromide (MTT) is reduced by enzymes in viable cells to form purple-colored formazan, thus reflecting the metabolic activity of the biofilms [24]. We mixed 50 μL of EcN bacterial suspension (approximately 10^8^ CFU/mL) with 50 μL of OA (128, 64, 32, and 16 μg/mL) separately, and then, the samples were added to a 96-well plate and incubated at 37 °C for 24 hours.The supernatant was discarded and washed twice with 100 μL PBS buffer, and MTT was dissolved in PBS solution to prepare a stock solution of 0.5 mg/mL. Then, 100 μL of MTT solution was added to each well and incubated in darkness for 3–4 h. The MTT solution was then aspirated and dissolved in 100 μL of DMSO, and the absorbance value was measured at OD_600nm_.

### 2.7. Extracellular Polysaccharide Determination

The ruthenium red staining method was used to determine the effect of OA on the extracellular polysaccharide matrix in EcN biofilms [25]. The ruthenium red solution was prepared as a 0.1% stock solution and diluted 10 times before use. An equal amount of EcN bacterial suspension (about 10^8^ CFU/mL) and OA was mixed in a 6-well plate. This was incubated for 24 h; then, the supernatant was discarded, washed twice with PBS buffer, and 1 mL of ruthenium red working solution was added to each well; 1 mL of ruthenium red solution was added as a blank in wells without biofilm. This was incubated in the dark for 1–2 h and then the liquid containing ruthenium red staining agent was absorbed into new 6-well plates. The absorbance value was measured at OD_530nm_. The OD value of the dye bound to the biofilm was equal to the blank group OD value minus the actual measured OD value of each sample well.

### 2.8. Viable Bacteria in Biofilms

Using a 6-well cell culture plate to detect the number of viable EcN biofilms after intervention with oleanolic acid, equal amounts of EcN bacterial suspension (about 10^8^ CFU/mL) and OA were mixed and added to the 6-well cell culture plate. After incubation for 24 h, 1 mL of a 0.1% Triton solution was added, fully dissolved, and enriched in the biofilm. The biofilm bacteria were quantitatively analyzed using the plate counting method.

### 2.9. Determination of Bacterial Surface Hydrophobicity

The affinity of microbial strains to hydrocarbons was used to reflect the surface hydrophobicity of the strains [26]. Fifteen milliliters of bacterial suspension were centrifuged at 8000 rpm for 10 min; then, the supernatant was discarded and the bacterial precipitate was resuspended in a PBS solution. The absorbance value (A_0_) was measured at OD_600nm_. OA was added to the EcN bacterial suspension and mixed with 1 mL of xylene. The blank group was prepared using an equal amount of PBS buffer. The mixture was left to stand at 37 °C for 1 h after being vortexed for 3 min. The aqueous phase was collected and the absorbance (A_1_) was measured at OD_600nm_. The formula for calculating surface hydrophobicity was as follows:Hydrophobicity (%) = (A_0_ − A_1_)/A_0_ × 100%(1)

Among them, A_0_ represents the initial absorbance of the bacterial body and A_1_ represents the final absorbance after xylene treatment.

### 2.10. Confocal Laser Scanning Microscopy (CLSM)

CLSM was used to observed changes in viable bacteria during biofilm formation [27]. Equal amounts of EcN bacterial suspension (approximately 10^8^ CFU/mL) and OA were added to the Lab TekTMII chamber cover glass and incubated at 37 °C for 24 h. The supernatant was discarded, washed twice with 0.9% NaCl solution, and stained with a FilmtracerTMIVE/DEADTM Biofilm Viability Kit for 20 min. Residual dyes were washed off with sterile water, and the morphology and structure of the biofilm were observed using laser confocal microscopy. The excitation wavelengths were set at 561 nm (PI) and 488 nm (SYTO), respectively.

### 2.11. Transcriptomic Analysis

#### 2.11.1. RNA Extraction, Library Construction, and RNA Sequencing

As described by the manufacturer, the total RNA of the OA intervention group and control group was separated using Trizol reagent (Invitrogen Life Technologies, Carlsbad, CA, USA), and then the concentration, mass, and integrity were measured using a NanoDrop spectrophotometer (Thermo Scientific, Waltham, MA, USA). Three micrograms of RNA were used as input material for the RNA sample preparations. Sequencing libraries were generated according to the following steps. First, mRNA was purified from total RNA using poly-T oligo-attached magnetic beads. Fragmentation was carried out using divalent cations under elevated temperature in an Illumina proprietary fragmentation buffer. First-strand cDNA was synthesized using random oligonucleotides and Super Script II. Second-strand cDNA synthesis was subsequently performed using DNA Polymerase I and RNase H. Remaining overhangs were converted into blunt ends via exonuclease/polymerase activities and the enzymes were removed. After adenylation of the 3′ ends of the DNA fragments, Illumina PE adapter oligonucleotides were ligated to prepare for hybridization. To select cDNA fragments of the preferred 400–500 bp in length, the library fragments were purified using the AMPure XP system (Beckman Coulter, Beverly, CA, USA). DNA fragments with ligated adaptor molecules on both ends were selectively enriched using the Illumina PCR Primer Cocktail in a 15 cycle PCR reaction. Products were purified (AMPure XP system, New England Biolabs, Ipswich, MA, USA) and quantified using the Agilent high sensitivity DNA assay on a Bioanalyzer 2100 system (Agilent, Santa Clara, CA, USA). The sequencing library was then sequenced on the NovaSeq 6000 platform (Illumina, San Diego, CA, USA).

#### 2.11.2. Raw Data Processing, Differential Gene Expression, and Functional Enrichment Analysis

Samples undergo sequencing on the platform to obtain image files, which are then processed by the sequencing platform’s software(Illumina Casava1.7 software used for basecalling), generating the original data in FASTQ format (raw data). Sequencing data may contain numerous connectors and low-quality reads. Therefore, we utilized fastp (0.22.0) software to filter the sequencing data, obtaining high-quality sequences (clean data) for subsequent analysis. We established a reference genome index using Bowtie2 and subsequently employed Bowtie2 to align the filtered reads to the ECN (NCBI, GCA_021559835.1) reference genome. HISAT2 (v2.1.0) was used to align the filtered reads to the reference genome.

We use HTSeq (v0.9.1) to statistically compare the read count values for each gene, representing the original gene expression. Subsequently, we standardized the expression using FPKM. Differential expression of genes was then analyzed using DESeq (v1.38.3), with the following screening conditions: absolute log2FoldChange > 1 and significant *p*-value < 0.05. Simultaneously, we utilized the R package Pheatmap (v1.0.12) to conduct bidirectional clustering analysis of all differentially expressed genes from the samples.

Employing topGO (v2.50.0) to conduct GO enrichment analysis on the differential genes (all DEGs/up DEGs/down DEGs), we calculated *p*-values using the hypergeometric distribution method. Significant enrichment was considered when *p*-value < 0.05. Subsequently, we identified the significantly enriched GO terms associated with differential genes to elucidate their main biological functions. ClusterProfiler (v4.6.0) software was employed to conduct enrichment analysis of the KEGG pathways associated with differential genes, emphasizing pathways with significant enrichment (*p*-value < 0.05). The Gene Set Enrichment Analysis (GSEA) (v4.1.0) tool was utilized for conducting gene set enrichment analysis of all genes, followed by the creation of a pathway map illustrating the enriched pathways.

### 2.12. qRT-PCR Analysis

To verify the reliability of the RNA Seq results, seven representative differentially expressed genes related to the biofilm were selected, and qRT-PCR was used to verify their consistency with transcriptomic differentially expressed genes. The gene primer sequence is shown in Appendix A, using *16SrRNA* as the housekeeping gene. Primer synthesis was completed by Tsingke Biotechnology Co., Ltd. (Beijing, China). According to the manufacturer’s instructions, reverse transcription was performed using the PrimeScript™ RT reagent kit with gDNA Eraser kit, and qRT-PCR was performed using the TB Green™ Premium EX Taq™ II (Tli RNaseH Plus, Baori Medical Biotechnology (Beijing) Co., Ltd., Beijing, China) kit. All amplifications were subjected to the following cyclic conditions: hot start at 95 °C for 30 s, followed by denaturation at 95 °C for 5 s, and annealing extension at 60 °C for 30 s. Dissolution curve analysis was performed using the ABI7500 Real-Time PCR system(applied biosystems, Waltham, MA, USA) at 60–95 °C. The relative mRNA levels of the tested genes were calculated using the 2^−ΔΔ Ct^ method.

### 2.13. Motility Assay

The swimming, twitching, and curling pilus movement phenotypes of EcN after OA intervention were tested using the method described by [28,29,30] and modified.

#### 2.13.1. Swimming Motility

A solid culture plate supplemented with 0.01 g/mL tryptone, 0.005 g/mL yeast extract, 0.005 g/mL sodium chloride, 0.003 g/mL bacterial agar powder, and 64 μg/mL OA was prepared. The culture plate without OA was used as a control; EcN was inoculated in the center of the plate and incubated at 37 °C for 48 h. Swimming motility was determined by measuring the bacterial motility diameter using a Vernier caliper.

#### 2.13.2. Twitching Motility

A solid culture plate supplemented with 0.01 g/mL tryptone, 0.005 g/mL yeast extract, 0.005 g/mL sodium chloride, 0.01 g/mL bacterial agar powder, and 64 μg/mL OA was prepared. The culture plate without OA was used as a control; EcN was inoculated in the center of the plate and incubated at 37 °C for 48 h. Twitching motility was determined by measuring the bacterial motility diameter using a Vernier caliper.

### 2.14. Congo Red Agar (CRA) Plate Assay

A 64 μg/mL OA solid culture plate supplemented with 0.0004 g/mL Congo Red, 0.005 g/mL sucrose, 0.0185 g/mL BHI medium, and 0.015 g/mL bacterial agar powder was prepared. The culture plate without OA was prepared as a control and 10 μL of EcN was inoculated in the center of the plate at 37 °C for 3–7 days. Colony morphology and size were observed.

### 2.15. Evaluation of the Effect of FeCl_3_ on EcN Biofilm

We chose to observe the effect of FeCl_3_ on the development of EcN biofilm, using the 96-well plate testing method and the method described by [31] with some modifications. Then, 128 μM FeCl_3_ solution was preconfigured and 256 μg/mL OA of MH medium was added to the 96-well plate in advance. FeCl_3_ was added in the same proportion and diluted in multiples to 32–1 μM. An equal amount of EcN bacterial suspension (about 10^8^ CFU/mL) was added and incubated at 37 °C for 24 h. After the biofilm was formed, quantitative staining with crystal violet was performed.

### 2.16. Statistical Analysis

Data were analyzed using GraphPad Prism 8.0 software. Student’s *t*-tests were used to calculate the statistical significance. Significant differences are indicated as * *p* < 0.05, ** *p* < 0.01, *** *p* < 0.001 and **** *p* < 0.0001. All experiments were conducted in multiple replicates, with a minimum of 3 replicates (*n* ≥ 3), and the data were expressed as mean ± standard deviation (SD).

## 3. Results

### 3.1. Determination of EcN Biofilm Formation Curve

EcN biomass formation at different time points was compared to determine the subsequent cultivation time and the results are shown in Figure 1b. The biomass of EcN was measured at 6, 18, 24, and 48 h, respectively. It was found that the adhesion stage of planktonic bacteria lasted for at least 6 h. By 18 h, the biofilm was in the initial adhesion state, exhibiting unstable membrane-forming ability. The biofilm at 24 h was more stable in adhesion compared to that at 48 h and exhibited a higher film-forming ability. Therefore, the subsequent cultivation conditions were all selected based on the 24 h biofilm.

### 3.2. Determination of Minimum Inhibitory Concentration of OA and Its Effect on EcN Growth

The antibacterial activity of OA was evaluated by the minimum inhibitory concentration (MIC) value. The MIC of OA to EcN was 128 μg/mL. The concentration of OA is higher than 128 μg/mL. At this concentration, the culture medium was clear, and no bacterial growth was observed. The concentration of OA was 64 μg/mL. The culture medium was turbid due to bacterial growth. As the solvent of OA was DMSO, and the MIC of DMSO to EcN was greater than 12.5%, the subsequent concentration of DMSO used in this experiment was 2%, which had no impact on the experimental results. The growth curves were investigated to identify the effects on growth of EcN due to OA exposure (Figure 1c). Research has found that OA does not affect the growth of EcN at 64, 32, 16, and 8 μg/mL levels.

### 3.3. OA Can Promote the Formation of EcN Biofilm and Have the Ability to Clear Biofilm against Pathogens

The effects of different concentrations of OA on the biofilm formation of EcN, *Salmonella* 011, and *Staphylococcus aureus* 002 were determined by crystal violet assay, as shown in Figure 1d–f. In Figure 1d, 8, 16, 32, and 64 μg/mL OA can all promote the formation of EcN biofilm, with biofilm growth percentages of 138.22%, 142.66%, 159.94%, and 178.08%, respectively. In fact, at higher concentrations of OA, the promoting effect on EcN biofilm formation is more pronounced.

When screening for active substances that promote the formation of biofilms in probiotics, it is also necessary to examine the inhibitory effect of these active substances on pathogenic bacteria. In view of this, the crystal violet staining method was used to determine that OA can inhibit the formation of biofilms of *Salmonella* 011 and *Staphylococcus aureus* 002, as shown in Figure 1e,f. After adding different concentrations of OA, the ability of *Salmonella* 011 *and Staphylococcus aureus* 002 to form biofilms was inhibited, with an average inhibition rate of 48.95% (*p* < 0.05) and 55.29% (*p* < 0.01), respectively.

### 3.4. The Effect of OA on the Metabolic Activity of EcN Biofilm Bacteria

The MTT method can preliminarily evaluate the effect of different concentrations of OA on the metabolic activity of EcN biofilm bacteria. Succinate dehydrogenase, located in the mitochondria of viable cells, has the ability to convert exogenous MTT into an insoluble blue-purple crystalline formazan, which is then accumulated within the cells. Conversely, non-viable cells do not possess this enzymatic activity. Dimethyl sulfoxide is used to solubilize the formazan within the cells, and its optical density is measured at a wavelength of 600 nm using a microplate reader. This quantification method serves as an indirect indicator of the quantity of viable cells present. In a defined range of cell concentrations, the extent of MTT crystallization is directly proportional to the cell count. A higher absorbance value corresponds to a greater number of viable bacteria at a given drug concentration, thereby indirectly reflecting the relative metabolic activity of the bacterial population. As shown in Figure 2a, under high-concentration OA intervention, the metabolic activity of biofilm bacteria increased by 152.83%. As the concentration of OA decreases, there is a dose-dependent relationship, with an average cellular metabolic activity of 72.81% (*p* < 0.001).

### 3.5. The Effect of OA on the Growth of Live Bacteria and Extracellular Polysaccharides during Biofilm Formation Process

Viable EcN in the biofilm was determined by plate counting. As shown in Figure 2b, the number of bacteria in the biofilm significantly increased after OA treatment (*p* < 0.0001). Compared with the control group, the number of viable bacteria in the biofilm treated with OA increased by 1.21 log10CFU/well.

The ruthenium red staining method was used to determine extracellular polysaccharides, as shown in Figure 2c. After OA intervention, the extracellular polysaccharides in the biofilm increased by 3.51 times compared to the control group (*p* < 0.0001).

### 3.6. The Effect of OA on the Surface Hydrophobicity of EcN

The surface hydrophobicity of EcN after OA treatment in this study is shown in Figure 2d. The results of the hydrophobicity test revealed the ability of OA to significantly increase the surface hydrophobicity of EcN. Compared with the control group, the hydrophobicity of the OA intervention group was 55.06% (*p* < 0.01).

### 3.7. Observation of Biofilm Morphology Using Confocal Laser Scanning Microscopy (CLSM)

The effect of OA on the formation of EcN biofilm was further studied using CLSM. Figure 3a–d show the representative orthogonal and 3D plots of the OA intervention on EcN biofilm. In the absence of OA, EcN forms a relatively loose biofilm with a lower thickness. In contrast, after OA treatment, EcN forms a dense biofilm. The number (Figure 3e), volume (Figure 3f), and surface area (Figure 3g) of EcN biofilms show significant differences (*p* < 0.001), with increases by 0.91 times, 4.39 times, and 0.93 times, respectively, while there is no significant difference in the base area of biofilms (Figure 3h). The total fluorescence intensity (Figure 3i) increased by 3.01 times, and SYTO staining (representing green live bacteria) and PI staining (representing red dead bacteria) were performed. The results are consistent with the quantitative results of crystal violet staining, indicating that OA has an impact on the ability and structure of EcN biofilm formation.

### 3.8. Transcriptomic Analysis

#### 3.8.1. Differential Expression Analysis

The statistical results of the quality evaluation of sequencing data are shown in Appendix A, indicating the reliability of the Illumina sequencing results. Compared with the control group, RNA sequencing analysis identified a total of 349 differentially expressed genes. The volcano plot of differentially expressed genes (Figure 4a) showed that 134 genes were upregulated (indicated in red) and 215 genes were downregulated (indicated in blue), with gray indicating insignificant differences. Cluster analysis was performed on the screened differentially expressed genes, and the clustering results showed that the genes were differentially expressed in the control group and OA group, with most genes showing the opposite trend (Figure 4b). In order to facilitate the sorting of useful information on the mechanism of OA’s effect on EcN biofilm formation, relevant information on differentially expressed genes in biofilms has been compiled. These genes may exert their mechanisms of action through the following pathways, including iron uptake system, adaptability factors, two-component system, c-di-GMP, stress, etc. The gene function classification is summarized in Table 1.

#### 3.8.2. KEGG Pathway Enrichment Analysis

KEGG analysis of differentially expressed genes related to biofilms showed that 22 pathways of differentially expressed genes (DEGs) were enriched after treatment with oleanolic acid. These pathways are mainly associated with metabolism, environmental processes, bacterial diseases, and cellular processes (Figure 4d). Several metabolic pathways, including the metabolism of terpenes and polyketides, as well as amino acid metabolism (lysine, arginine), and environmental process pathways such as signal transduction and bacterial movement in cellular processes, are significantly enriched.

Further analysis indicates that oleanolic acid treatment alters a series of important physiological processes, which may be related to bacterial motility, the QS system, iron uptake system, two-component system, and amino acid metabolism. Among them, upregulated differentially expressed genes (DEGs) are significantly enriched in the transcriptional regulation of iron uptake and dual-component systems, while downregulated genes mainly enrich the biosynthesis of flagella and flagella assembly.

#### 3.8.3. GO Enrichment Analysis

To further determine the molecular characteristics of Differentially Expressed Genes (DEGs), functional classification was performed using Gene Ontology (GO) enrichment analysis. As shown in Figure 4e, the top 30 items with the highest enrichment mainly consist of biological processes, such as iron carrier transport, nucleic acid template transcription regulation, RNA metabolism, and biosynthesis regulation. In the molecular functional category, Differentially Expressed Genes (DEGs) typically belong to catalytic activity, binding enzymes, and transport proteins. The biological processes involve pilus synthesis, cell outer membrane interaction, and interaction with enterocolin synthase complexes.

### 3.9. QRT PCR Validation

To verify the reliability of the RNA-Seq results, seven genes related to the biofilm were randomly selected for qRT-PCR validation. Three genes were upregulated, and four genes were downregulated. The qRT-PCR validation results were consistent with the RNA-Seq results (Figure 4c), indicating the reliability of the transcription results.

### 3.10. Perform Relevant Phenotypic Tests on Transcriptomic Results

#### 3.10.1. OA Affects the Exercise Phenotype of EcN

It is known that flagella and pili are involved in bacterial swimming motility and twitching motility, playing important roles in biofilms [32,33,34]. In the presence of OA, genes related to flagella and pili are significantly downregulated (Table 1). To further validate the transcriptomic analysis results, it was determined that OA plays a role in swimming motility and twitching motility associated with biofilm formation. Interestingly, OA exhibited a distinctive exercise phenotype on EcN. Following OA intervention, it decreased by 36.23% in the diameter of biofilm bacteria compared to the control group, as illustrated in Figure 5b. During twitching motility, it decreased by 51.13% compared to the control group (Figure 5a). The presence of OA in agar plates of various concentrations significantly reduced the swimming and twitching motility of EcN (*p* < 0.001).

#### 3.10.2. OA Affects the Production of EcN Curli

The culture that produced dark red colonies was considered to express curli by assessing the production of curli pili using Congo red agar plates (Figure 5c). After the OA intervention, the colony diameter decreased by 31.11% compared to the control group (*p* < 0.001). However, there was no significant change in color. The alteration in morphology and size of curli indicated that curli was still being expressed after the addition of OA. However, the expression level decreased instead of being inhibited, possibly due to the regulation of a specific gene.

#### 3.10.3. FeCl_3_ Promotes the Formation of EcN Biofilm

The impact of various concentrations of FeCl_3_ on EcN biofilm was examined on a polyethylene 96-well plate (Figure 5d). It was observed that there was no significant difference when FeCl_3_ alone was introduced to the EcN biofilm. At 2 μM, the co-action of FeCl_3_ and culture medium containing OA resulted in significantly more biofilm formation in EcN compared to culture medium containing only OA (*p* < 0.01).

## 4. Discussion

Probiotics can improve the microecological balance of the body and have special effects in promoting nutrient absorption, controlling intestinal infections, and regulating immune function. In this study, traditional Chinese medicine active substances were screened as inducers of EcN biofilm formation using the biofilm quantification method (Appendix A). It was found that the effect of oleanolic acid was most obvious for promoting the formation of EcN biofilms. By measuring the extracellular polysaccharides, viable bacterial count, metabolic activity, and surface hydrophobicity of biofilm bacteria, as well as characterizing bacterial morphology using laser confocal microscopy, the impact of oleanolic acid on EcN biofilm formation was thoroughly assessed. However, its mechanism remains largely unknown. Thus, the effect mechanism of OA on ECN was examined using transcriptomics, and the results indicated that it related to the fimbriae, flagella, and iron transport.

Type I pili are encoded by fim gene clusters located on chromosomes, with FimA as the primary structural subunit. Type I pili have been shown to be necessary for *E. coli* to adhere to non-biological surfaces [35]. Research has found that the amino acid mutation at the FimH site of *Salmonella typhimurium* is advantageous for adhesion to HEp-2 cells, leading to the capacity to form biofilms [36]. However, some studies have also shown that type I pili have a negative or no regulatory effect on biofilms [37]. Overall, the regulation of biofilms by type I pili involves the coordinated regulation of multiple genes. The expression of type I pili is associated with bacterial adhesion and the regulation of certain outer membrane proteins, such as OmpA, Slp, and OmpX [38]. Our study found that the gene *fimA*, related to type I pili synthesis, was significantly downregulated. Combined with phenotype validation analysis, it is speculated that the action of oleanolic acid reduces bacterial motility and enhances initial biofilm adhesion. The mechanism of action also requires knocking out and repairing the *fimA* gene to confirm it as the key target of action.

Curled pili primarily enhance adhesion and colonization ability by binding to fibronectin and laminin and producing amyloid-like proteins. The formation of *Escherichia coli* biofilm is closely related to these functions, which mainly include adhesion to epithelial cells, enhancing cell automatic aggregation ability, and increasing hydrophobicity on bacterial surfaces [39]. CsgD is the primary regulatory factor for the biosynthesis of curli pili and cellulose. It can positively regulate the *csgBA* promoter, but its activity is also influenced by other regulatory factors [40]. The expression of the CsgD protein leads to transcriptional changes in at least 24 new genes. However, *cpxA/R* and *rcs* can inhibit the transcription and expression of *csgD*, leading to a decrease in Curli pili synthesis [40]. FecR is a regulatory protein involved in iron metabolism. Inhibiting the *fecR* gene could potentially amplify the positive impact of CsgD on biofilm formation. The iron-sensitive genes (*fecR* and *fhuE*) in CsgD regulators are primarily utilized to regulate the intracellular iron concentration during the transition from planktonic bacteria to adherent bacteria [41]. It is speculated that the overexpression of *fecR* leads to an increase in iron regulatory proteins in the bacterial body, indirectly resulting in a decrease in CsgD expression. There may be negative regulation between the two. In summary, it is speculated that intervention with oleanolic acid may weaken the expression of CsgD, whether it is the dual-component system or the iron uptake system activated by the body’s response. This intervention will regulate CsgD and reduce pili synthesis, thereby affecting the formation of biofilms. Based on preliminary phenotypic verification, it is speculated that oleanolic acid may affect the synthesis of EcN curli pili through different pathways. There are also studies indicating that the CSG region encodes pili-related genes. *csgD* controls cell aggregation and inhibits flagella synthesis by regulating Curli pili [42].

The regulation of flagella is controlled by multiple systems, including dual-component systems and c-di-GMP levels. DgcZ is the primary cyclic dimer diguanylate cyclase (DGC) responsible for producing GMP (c-di-GMP), which in turn stimulates the biosynthesis of PGA. The transcription of dgcZ is responsive to the lipoprotein NlpE and activates the Cpx two-component system. CpxR inhibits bacterial motility and stimulates the transcription of dgcZ. The fumarate reductase (FRD) complex plays a role in regulating flagella assembly, enhancing the expression of dgcZ, and mediating biofilm formation, which is essential for bacterial adhesion to surfaces [43]. High c-di-GMP content was found in *Pseudomonas aeruginosa* PA-HN002, which led to reduced bacterial motility and enhanced biofilm formation ability [44]. This study found an upregulation of dgcZ related to GMP synthesis. This suggests that after the action of oleanolic acid, the expression level of dgcZ increases, activating the two-component system in a c-di-GMP-dependent manner. This intervention affects flagella assembly, inhibiting motility, and promoting biofilm formation through this regulatory mechanism. The specific regulatory mechanism still needs to be validated by knocking out the dgcZ gene and measuring the content of c-di-GMP. Due to changes in the characteristics of bacterial flagella and flagella after intervention with oleanolic acid, it affects the initial adhesion of bacteria. Preliminary experiments have investigated the surface hydrophobicity of bacteria to indirectly reflect their adhesion characteristics.

Iron transporters also contributed considerably to the formation of biofilms when OA treated the ECN. Moderate amounts of iron are beneficial for the formation of biofilms in *Bacillus subtilis*, *Pseudomonas aeruginosa*, *Escherichia coli*, and *Staphylococcus aureus* [45,46,47]. Iron can promote the formation of biofilms in *Bacillus subtilis* through two main mechanisms: trivalent iron facilitates electron transfer and generates a strong membrane potential during aerobic respiration, and it can also bind to the extracellular matrix of biofilms [48]. It is speculated that EcN also has a similar effect. After the action of oleanolic acid, it is beneficial for trivalent iron to tightly bind to the extracellular matrix of EcN biofilm, thereby promoting biofilm formation. Meanwhile, it should be noted that different concentrations of iron can affect the characteristics of pili and pore proteins, thereby further influencing the virulence of the bacterial body. Therefore, when ingesting high concentrations of iron, the toxic effects on the body should also be considered [49]. When the balance of iron homeostasis undergoes subtle changes, it may lead to the inhibition of bacterial growth and redox stress. Importantly, iron chelation can prevent oxidative stress by inhibiting the Fenton reaction [45]. This study found that after intervention with oleanolic acid, multiple genes in the iron uptake system were significantly upregulated. Some studies have shown that even if the concentration of iron is 10 times higher in *Pseudomonas putida* (182 μM), it can still inhibit bacterial swarm movement [50]. This indicates that iron concentration at a certain level will inhibit swarm movement. Exogenous addition of excessive iron to *Pseudomonas aeruginosa* can inhibit bacterial swarm movement, accompanied by the expression of rhlA [51]. In addition, the iron uptake process of *Bacillus parahaemolyticus* is closely related to flagellar movement [52]. In summary, it is speculated preliminarily that after the action of oleanolic acid on EcN, the same mechanism of action also exists. When iron uptake increases, bacterial motility weakens, resulting in enhanced initial adhesion ability and promoting the formation of EcN biofilm. Preliminary evidence suggests that oleanolic acid affects iron uptake and participates in the formation of EcN biofilms.

In addition, it is worth noting that verifying the successful colonization of probiotics in the form of biofilms requires studies to be conducted in appropriate in vivo models. These studies should evaluate not only the establishment and formation of probiotic biofilms in the intestine but also how the administration of oleanolic acid can impact the intestinal microbiota, as well as metabolic and immunological aspects both locally and systemically in the host interaction.

## 5. Conclusions

This study found that oleanolic acid not only significantly promotes the formation of EcN biofilm but also inhibits the formation of pathogenic bacterial biofilm. Based on transcriptomic results and subsequent phenotype validation analysis, it is preliminarily indicated that oleanolic acid primarily inhibits bacterial movement, enhances initial adhesion, and promotes the formation of EcN biofilm. Additionally, by regulating iron uptake, it interacts with oleanolic acid within an appropriate concentration range to jointly regulate the formation of EcN biofilm. This study lays the theoretical foundation for the development of new and effective probiotic products for intestinal colonization.

## Figures and Tables

**Figure 1 microorganisms-12-01097-f001:**
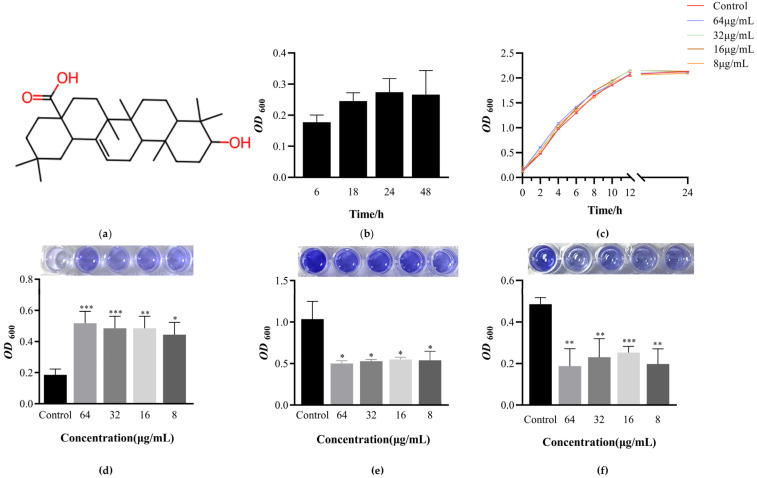
The effect of oleanolic acid on the formation of biofilms of EcN, *Salmonella*, and *Staphylococcus aureus*. (**a**) The chemical structural formula of oleanolic acid. (**b**) EcN biofilm formation curve. Biomass was determined by crystal violet staining and measuring absorbance at OD_600nm_. (**c**) The effect of different concentrations of oleanolic acid on the growth of EcN. Bacterial growth detected by measuring the optical density at OD_600nm_. (**d**) The effect of oleanolic acid on the EcN biomass. (**e**) The effect of oleanolic acid on the *Salmonella* biomass. (**f**) The effect of oleanolic acid on the *Staphylococcus aureus* biomass. In all panels, bars represent the average of at least three biological replicates. The error line represents the standard deviation. Unpaired *t*-test (two-tailed) was used to measure statistical significance. * *p* < 0.05, ** *p* < 0.01 and *** *p* < 0.001 compared with the control group.

**Figure 2 microorganisms-12-01097-f002:**
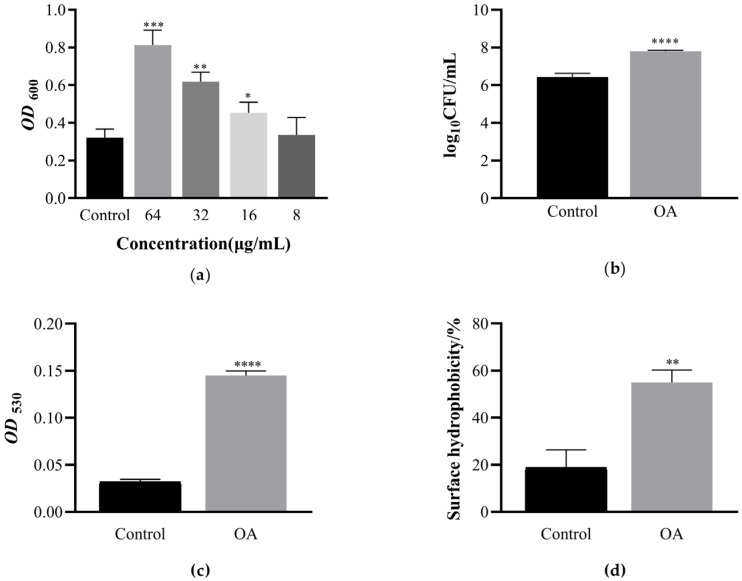
Oleanolic acid affects the bacterial metabolic activity, viable count, extracellular polysaccharide content, and surface hydrophobicity of EcN biofilms. (**a**) The effect of oleanolic acid on the metabolic activity of EcN biofilm bacteria. Bacterial metabolic activity measured using thiazole blue colorimetry and absorbance at OD_600nm_. (**b**) The effect of oleanolic acid on the number of viable bacteria in EcN biofilm. Utilizing the colony plate counting method to determine the quantity of viable bacteria in a biofilm. (**c**) The effect of oleanolic acid on EcN extracellular polysaccharides. Extracellular polysaccharide content of EcN biofilm measured using the ruthenium red staining method and the absorbance was measured at OD_530nm_. (**d**) The effect of oleanolic acid on the surface hydrophobicity of EcN. Determination of the surface hydrophobicity of EcN based on its varying affinities to hydrocarbons. In all panels, bars represent the average of at least three biological replicates. The error line represents the standard deviation. Unpaired *t*-test (two-tailed) was used to measure statistical significance. * *p* < 0.05, ** *p* < 0.01, *** *p* < 0.001 and **** *p* < 0.0001 compared with the control group.

**Figure 3 microorganisms-12-01097-f003:**
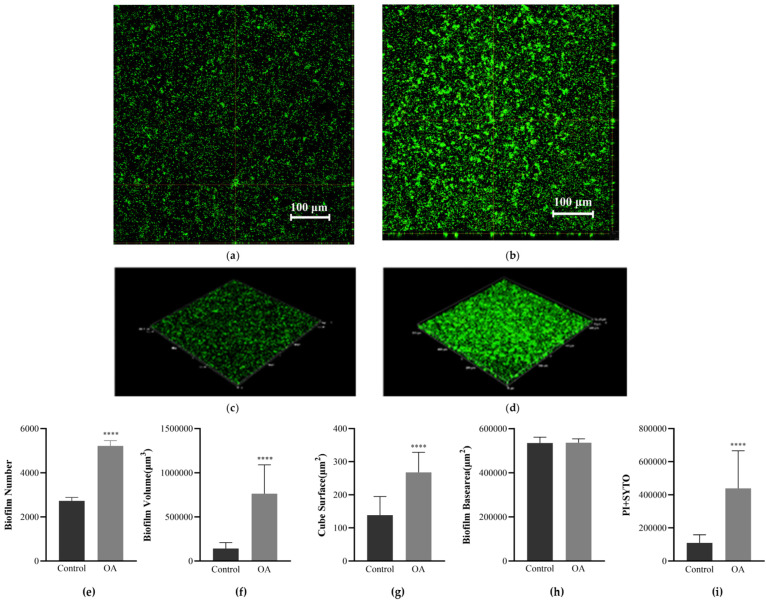
CLSM observation of the effect of oleanolic acid on EcN biofilm formation. (**a**) Control representative orthogonal diagram. (**b**) Representative orthogonal diagram of intervention with oleanolic acid in EcN. (**c**) Control representative 3D diagram. (**d**) Representative 3D diagram of intervention with oleanolic acid in EcN. (**e**) Number of biofilms. (**f**) Biofilm volume. (**g**) Surface area of biofilm. (**h**) Base area of biofilm. (**i**) Total fluorescence intensity (SYTO + PI). The excitation wavelengths are 561 nm (PI) and 488 nm (SYTO), respectively. Three independent repeated trials were conducted, with randomly selected fields of view for imaging. At least 12 field of view images were selected for the analysis of biofilm-related parameters. Unpaired *t*-test (two-tailed) was used to measure statistical significance. **** *p* < 0.0001 compared with the control group.

**Figure 4 microorganisms-12-01097-f004:**
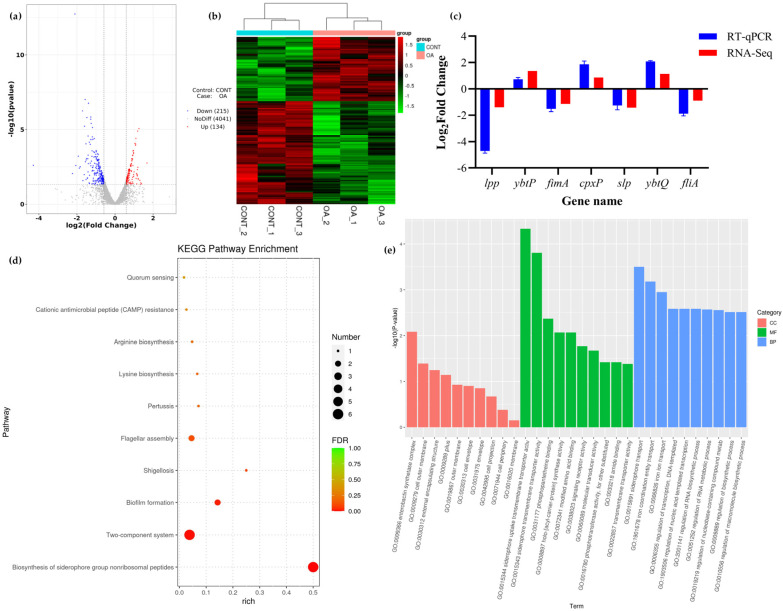
Transcriptome data analysis of oleanolic acid intervention on EcN biofilm. (**a**) Volcano map showing differential expression. (**b**) Chart for cluster analysis. (**c**) Comparison of differentially expressed genes between qRT-PCR and RNA-Seq. (**d**) KEGG enrichment statistical map of differentially expressed genes associated with biofilm. (**e**) GO enrichment results of differentially expressed genes linked to biofilm.

**Figure 5 microorganisms-12-01097-f005:**
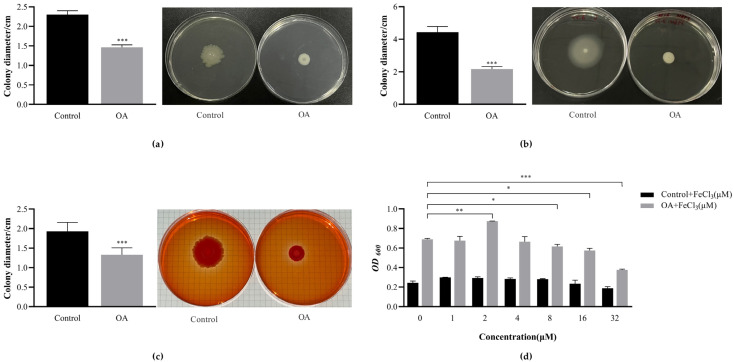
Relevant phenotypic tests performed for transcriptomic results. (**a**) The effect of oleanolic acid on the bacterial twitching movement. Observation of twitching motility diameter using a 1% agar diffusion test. (**b**) The effect of oleanolic acid on the bacterial swimming movement. Observation of swimming motility diameter using a 0.3% agar diffusion test. (**c**) Observation of colony morphology on Congo red agar plates. (**d**) The effect of FeCl_3_ on the formation of EcN biofilm. The biomass of FeCl_3_ and the combined effect of FeCl_3_ and oleanolic acid on EcN using the crystal violet staining method were determined. The absorbance was measured at OD_600nm_. In all panels, bars represent the average of at least three biological replicates. The error line represents the standard deviation. * *p* < 0.05, ** *p* < 0.01 and *** *p* < 0.001 compared with the control group.

**Table 1 microorganisms-12-01097-t001:** Differentially expressed genes related to biofilms.

Gene ID	Gene	Product	Log2FC	*p*-Value
Iron uptake system				
RS09490	*irp2*	yersiniabactin biosynthetic protein	1.2530	8.66 × 10^−6^
RS09500	*ybtP*	yersiniabactin ABC transporter ATP-binding/permease protein YbtP	1.3492	3.75 × 10^−2^
RS09505	*ybtQ*	yersiniabactin ABC transporter ATP-binding/permease protein YbtQ	1.1286	8.81 × 10^−3^
RS17060	*entH*	proofreading thioesterase in enterobactin biosynthesis	1.2178	2.09 × 10^−2^
RS20935	*fecR*	ferric citrate regulator FecR	1.1570	1.10 × 10^−2^
RS11870	*dmsB*	dimethyl sulfoxide reductase subunit B	0.7426	1.54 × 10^−2^
RS17130	*fepA*	ferric enterobactin outer membrane transporter	0.7580	1.15 × 10^−2^
RS17095	*fepD*	ferric enterobactin outer membrane transporter	0.6541	2.15 × 10^−2^
RS14160	*fhuE*	ferric coprogen/ferric rhodotorulic acid outer membrane transporter	0.8396	2.32 × 10^−3^
RS16085	*fiu*	iron catecholate outer membrane transporter Fiu	0.7884	5.44 × 10^−3^
RS09515	*ybtS*	salicylate synthase Irp9	0.9727	1.39 × 10^−2^
RS17135	*entD*	phosphopantetheinyl transferase	0.8360	2.92 × 10^−3^
RS17075	*entE*	2,3-dihydroxybenzoate-AMP ligase	0.7360	2.35 × 10^−2^
RS09485	*irp1*	yersiniabactin polyketide synthase HMWP1	0.8076	2.83 × 10^−3^
RS17115	*entF*	activating enzyme	0.8829	4.21 × 10^−2^
Adaptability factors				
RS12115	*fimA*	major type 1 subunit fimbrin	−1.1391	1.79 × 10^−2^
RS11230	*ihfA*	integration host factor subunit alpha	−1.0739	1.80 × 10^−3^
RS14465	*csgE*	curli assembly component CsgE	−1.0947	9.19 × 10^−3^
RS14460	*csgD*	transcriptional regulator	−0.9228	1.11 × 10^−3^
RS18375	*ecpA*	common pilus major subunit	−0.6306	2.49 × 10^−2^
RS18370	*ecpR*	transcriptional regulator for the ecp operon	−0.7083	9.96 × 10^−3^
RS01085	*hdeD*	acid-resistance membrane protein	−1.0482	6.51 × 10^−4^
RS14310	*flgM*	anti-sigma factor for FliA	−0.9128	2.32 × 10^−2^
RS09885	*fliA*	RNA polymerase sigma factor FliA	−0.9049	8.27 × 10^−3^
RS18390	*ecpD*	polymerized tip adhesin of ECP fibers	0.6233	2.99 × 10^−2^
RS01150	*slp*	outer membrane lipoprotein	−1.4296	8.88 × 10^−4^
RS13225	*hns*	DNA-binding transcriptional dual regulator H-NS	−0.8428	1.99 × 10^−3^
Two component system				
RS06395	*qseE/GlrK*	two component system sensor histidine kinase QseE/GlrK	0.7615	5.53 × 10^−3^
RS13670	*ymgA*	putative two-component system connector protein YmgA	0.9766	1.06 × 10^−2^
RS23010	*cpxP*	inhibitor of the cpx response periplasmic adaptor protein	0.8645	8.94 × 10^−4^
RS01145	*dctR/yhiF*	LuxR family repressor for dicarboxylate transport	−1.0504	2.56 × 10^−2^
Stress				
RS19265	*degP*	periplasmic serine endoprotease DegP	0.5917	2.83 × 10^−2^
c-di-GMP				
RS01915	*argD*	N-acetylornithine aminotransferase/N-succinyldiaminopimelate aminotransferase	−1.0445	4.18 × 10^−2^
RS11960	*dgcZ*	diguanylate cyclase	0.9421	4.09 × 10^−2^

## Data Availability

The original contributions presented in the study are included in the article/Appendix A, further inquiries can be directed to the corresponding author.

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
