# Peer review of "Oleanolic Acid Promotes the Formation of Probiotic Escherichia coli Nissle 1917 (EcN) Biofilm by Inhibiting Bacterial Motility"

_microorganisms, 2024, doi:10.3390/microorganisms12061097_

Round 1

Reviewer 1 Report

Comments and Suggestions for Authors

Sections 2.2-2.7,  2.10 and 2.13-2.15 are written in their entirety or in parts as protocols with instructions rather than a report on what has been done and how. There should be a change of verbal mode from imperative to simple past.

Section 2.3: please clarify the concentrations of oleanolic acid and how they were chosen.

The source and origin of EcN, Salmonella 011,  and Staphylococcus aureus 002 have not been declared in section 2. The spectrophotometer(s) used have also not been identified.

“Fifteen milliliters of bacterial suspension was centrifuged…”. Please make the agreement between the verb and subject number (plural for both or singular for both). Similar disagreements were also found throughout the paper in other sentences.

“Ffteen milliliters of bacterial suspension was centrifuged at 8000 rpm for 10 min, and the supernatant was discarded. The absorbance value (A0) was measured at OD 600nm  using a PBS solution.” It is hard to understand how it was worked here. Was the value absorbance measured on the supernatant or on the precipitate? What was the role of the PBS solution? How valid are the measurements of a suspension absorbance (if we understand correct that measurements were performed on a suspension –“the bacterial body”).

Line 228: why are quotations marks placed at the end of this sentence?

Sections 2.13.1 and 2.13.2: please clarify what tool was used to measure the bacterial movement diameters.

“One-way analysis of variance was used, followed by multiple tests to calculate the  statistical significance of differences.” “Multiple tests” is very vague. If the authors used post-hoc tests specific for ANOVA (or FDR) they should specifically clarify what tests they used.

“All experiments were conducted in triplicate (n ≥ 3)”. If experiments were conducted in triplicate, then n=3. If n ≥ 3, it should be clarified that experiments were conducted in multiple replicates, minimum 3.

Figure 1: please explain in the caption or in the text why the control values differ so widely among the three plots (letters d-f).

Lines 452-461: This text could/should be accompanied by a few relevant references (particularly where it is mentioned that “there are numerous reports…”).

Lines 603-604: it seems likely that an automatic translator is responsible for this, but there is no such thing as a “bacterial bee movement”. As the cited reference suggests, probably the authors intended “swarm movement”.

Comments on the Quality of English Language

Moderate English language editing is necessary.

Author Response

Dear expert, thanks for your valuable comments and suggestions, which really helped us to improve this paper. All the responses associated with your comments and questions were described as below.

Reviewer 1:

1.Sections 2.2-2.7, 2.10 and 2.13-2.15 are written in their entirety or in parts as protocols with instructions rather than a report on what has been done and how. There should be a change of verbal mode from imperative to simple past.

Response:Thank you for your comment. We have carefully corrected the grammar errors in the manuscript of the above chapters using revision mode.

2.Section 2.3: please clarify the concentrations of oleanolic acid and how they were chosen.

Response: Thanks for your comments. It has been added to the Section 2.3 section. 

The concentration of the OA reserve solution here is 8192 μg/mL. Choose the 13th power of 2, dilute it proportionally and sequentially, and observe the minimum inhibitory concentration of OA on EcN.

3.The source and origin of EcN, Salmonella 011, and Staphylococcus aureus 002 have not been declared in section 2. The spectrophotometer(s) used have also not been identified.

Response: Thanks for your comments. It has been added to the Section 2.1 section according to your suggestion and highlighted in red. As shown below:

Oleanolic acid (OA)with a purity of 97% was stored at 4°C and was purchased from Shanghai McLean Biochemical Technology Co., Ltd. The OA reserve solution in this experiment was dissolved in dimethyl sulfoxide. Escherichia coli Nissle 1917 purchased from the China General Microbiological Culture Collection Center. EcN was inoculated into MH medium and incubated on a shaking table for 5-6 hours to reach the logarithmic growth phase.while Salmonella 011 from pigs and Staphylococcus aureus 002 from sheep were isolated and preserved in the College of Veterinary Medicine at Southwest University. The strains were stored at -80°C in 60% glycerol broth for seed preservation. Separated and cultivated bacterial strains by streaking them on agar plates into three zones, and stored them at 4°C. Used each agar plate about a week to maintain its bacterial vitality.The UV visible spectrophotometer was purchased from Hanyi Instrument (Shanghai) Co., Ltd

4.“Fifteen milliliters of bacterial suspension was centrifuged…”. Please make the agreement between the verb and subject number (plural for both or singular for both). Similar disagreements were also found throughout the paper in other sentences.

Response: Thanks for your comments. We have carefully corrected the grammar errors in this manuscript using revision mode, and the errors have been highlighted in red after revision. As shown below:

Fifteen milliliters of bacterial suspension were centrifuged at 8000 rpm for 10 min, and the supernatant was discarded.

5.“Ffteen milliliters of bacterial suspension was centrifuged at 8000 rpm for 10 min, and the supernatant was discarded. The absorbance value (A0) was measured at OD 600nm using a PBS solution.” It is hard to understand how it was worked here. Was the value absorbance measured on the supernatant or on the precipitate? What was the role of the PBS solution? How valid are the measurements of a suspension absorbance (if we understand correct that measurements were performed on a suspension –“the bacterial body”).

Response: Thanks for your comments. We have carefully reviewed the content and found that some expressions were incorrect. After revision, they were highlighted in red. As shown below:

Fifteen milliliters of bacterial suspension were centrifuged at 8000 rpm for 10 min, and discard the supernatant.and resuspend the bacterial precipitate in a PBS solution.The absorbance value (A0) was measured at OD600nm.

Centrifuge the bacterial suspension at 8000 rpm for 10 minutes, discard the supernatant, and resuspend the bacterial precipitate with a PBS solution. The absorbance value is measured in the bacterial suspension containing PBS. The function of PBS is to replace the broth in the bacterial suspension. The broth itself affects the absorbance value and also prevents bacterial regrowth. Xylene was introduced to facilitate the separation of the organic phase from the aqueous phase. Following a one-hour incubation period, the aqueous phase was retrieved. During this interval, the bacterial sinked slowly, and subsequent absorbance readings were taken post re-vortexing.

6.Line 228: why are quotations marks placed at the end of this sentence?

Sections 2.13.1 and 2.13.2: please clarify what tool was used to measure the bacterial movement diameters.

Response: Thanks for your comments. The punctuation used here is incorrect and has been revised and removed. The tool for measuring bacterial movement diameter has been added to Sections 2.13.1 and 2.13.2. After revision, they were highlighted in red. As shown below:

2.13.1 Swimming motility

Prepared a solid culture plate supplemented with 0.01 g/mL tryptone, 0.005 g/mL yeast extract, 0.005 g/mL sodium chloride, 0.003 g/mL bacterial agar powder, and 64 μg/mL OA. Used the culture plate without OA as a control, inoculated EcN in the center of the plate, incubated at 37°C for 48 hours, and determined swimming motility by measuring the bacterial motility diameter using a Vernier caliper.

2.13.2 Twitching motility

Prepared a solid culture plate supplemented with 0.01 g/mL tryptone, 0.005 g/mL yeast extract, 0.005 g/mL sodium chloride, 0.01 g/mL bacterial agar powder, and 64 μg/mL OA. Used the culture plate without OA as a control, inoculated EcN in the center of the plate, incubate at 37°C for 48 hours, and determined twitching motility by measuring the bacterial motility diameter using a Vernier caliper.

7.“One-way analysis of variance was used, followed by multiple tests to calculate the statistical significance of differences.” “Multiple tests” is very vague. If the authors used post-hoc tests specific for ANOVA (or FDR) they should specifically clarify what tests they used. 

Response: Thanks for your comments. We carefully checked the statistical methods and made revisions. As shown below:

Data were analyzed by GraphPad Prism 8.0 software. Student’s t-tests were used to calculate the statistical signifcance. Signifcant diferences are indicated as *p < 0.05, **p < 0.01, ***p < 0.001 and ****p < 0.0001

8.“All experiments were conducted in triplicate (n ≥ 3)”. If experiments were conducted in triplicate, then n=3. If n ≥ 3, it should be clarified that experiments were conducted in multiple replicates, minimum 3.

Response: Thanks for your comments. The errors here have been revised and highlighted in red. As shown below:

All experiments were conducted in multiple replicates, with a minimum of 3 replicates (n ≥ 3), and the data were with data expressed as mean ± standard deviation (SD).

9.Figure 1: please explain in the caption or in the text why the control values differ so widely among the three plots (letters d-f). 

Response: Thanks for your comments. The difference in d-f control values is due to the fact that these are three different bacteria, namely EcN, Salmonella, and Staphylococcus aureus, and their ability to form biofilms varies. The meaning of the symbol d-f was clearly indicated in the accompanying caption.

(d) The effect of oleanolic acid on the EcN biomass;(e) The effect of oleanolic acid on the Salmonella biomass; (f) The effect of oleanolic acid on the Staphylococcus aureus biomass.

10.Lines 452-461: This text could/should be accompanied by a few relevant references (particularly where it is mentioned that “there are numerous reports…”).

Response: Thanks for your comments. Corresponding references have been added.

11.Lines 603-604: it seems likely that an automatic translator is responsible for this, but there is no such thing as a “bacterial bee movement”. As the cited reference suggests, probably the authors intended “swarm movement”.

Response: Thanks for your comments. The description here is indeed about swarm movement, and we have carefully revised it. After revision, they were highlighted in red. As shown below:

Some studies have shown that even if the concentration of iron is 10 times higher in Pseudomonas putida (182 μM), it can still inhibit bacterial swarm movement [59]. This indicates that iron concentration at a certain level will inhibit swarm movement. Exogenous addition of excessive iron to Pseudomonas aeruginosa can inhibit bacterial swarm movement, accompanied by the expression of rhlA [60].

Reviewer 2 Report

Comments and Suggestions for Authors

Dear Authors,

The manuscript “Oleanolic acid promotes the formation of probiotic Escherichia coli Nissle 1917 (EcN) biofilm by inhibiting bacterial motility” investigates the effect of oleanolic acid (OA) on the formation of biofilm by probiotic Escherichia coli Nissle 1917 (EcN) and explores the underlying mechanisms. The results demonstrate that OA promotes the formation of EcN biofilm by inhibiting bacterial motility. This finding suggests a potential application of OA in enhancing the probiotic properties of EcN. The study sheds light on the intricate relationship between dietary compounds and gut microbiota, providing insights into the modulation of microbial behavior for therapeutic purposes.

Abstract:

The abstract requires revisions to improve clarity and relevance. It lacks specificity in describing experimental methods and the observed effects. Key findings should be summarized concisely, specifying the concentration range of OA tested and the assessment techniques used. Additional insights from the study should also be highlighted. Furthermore, discussing potential clinical implications or applications would emphasize the research's significance in probiotic therapy and gut health.

Introduction

The introduction presents a broad overview of probiotics and biofilms, introducing the study's focus on probiotic Escherichia coli Nissle 1917 (EcN) biofilms and the potential role of oleanolic acid (OA). However, there are areas for improvement:

-       The introduction could benefit from a clearer structure, breaking down key concepts into distinct paragraphs. This would guide the reader through topics such as the definition and importance of probiotics, challenges in their application, the significance of biofilms, and the rationale for studying OA.

-       The transition from discussing probiotics generally to introducing EcN biofilms and OA is somewhat abrupt. A smoother transition could be achieved by clearly stating the research gap or question the study addresses, such as the need for novel approaches to enhance probiotic colonization in the gut.

-       While background information on biofilms is provided, more specific context related to EcN biofilms and OA's potential application is needed. Detailing why EcN biofilms are of interest and how OA fits into probiotic research would strengthen the introduction.

-       While the introduction cites studies to support various points, some claims could benefit from more direct citations or specific examples. Additionally, providing recent or relevant references to support the discussion on EcN biofilms and OA's effects would enhance credibility.

-       The study's objectives are briefly mentioned, but they could be articulated more clearly to highlight the specific research gap and aims of the investigation. Clearly stating research questions or hypotheses would provide a stronger framework for understanding the study's significance.

2.1 Materials and Methods - outlines the materials and methods used in the study but lacks specificity in several areas:

-       Details on the purity and handling/storage instructions of oleanolic acid (OA) are missing. Including information on the grade, purity, and handling requirements would ensure transparency and reproducibility.

-       More details on the culture media and growth conditions for maintaining and subculturing bacterial strains, such as Escherichia coli Nissle 1917 (EcN), are necessary.

-       Information on how microbial strains were stored and handled to maintain viability and purity should be included.

2.2 Determination of EcN biofilm formation curve - describes the methodology used to assess biofilm formation but lacks some crucial details: 

-       Specific details on culture medium, incubation time, and conditions are needed.

-       More information on the volume of bacterial suspension, washing steps, and specifics of the acetic acid solution used is necessary.

-       Details on controls and replicates used in the assay are essential for assessing the reliability of results.

-       Information on data processing and statistical analyses performed is missing.

2.3. Determination of minimal inhibitory concentration - outlines the methodology used to determine MIC but needs further clarification:

-       Briefly explaining the microdilution method would be helpful.

-       Details on OA solution volume, concentration range, and final well volume are needed.

-       Clarifying incubation conditions, such as covering the plate or agitation, is necessary.

-       Criteria for bacterial growth inhibition and interpretation of MIC values should be included.

-       Details on controls and replicates used in the experiment should be provided.

2.4. Growth curves - outlines the methodology used to assess OA's effect on bacterial growth but needs clarification:

-       Specifics on culture medium and growth conditions are necessary.

-       Rationale behind OA concentrations and their relevance should be included.

-       Details on sampling volume and uniform mixing are needed.

-       Information on data analysis methods and statistical analyses performed should be included.

-       Details on controls and replicates used are essential for assessing result reliability.

2.5 Biofilm formation - describes the methodology used to assess OA's impact on biofilm formation but requires further detail:

-       Explanation of OA concentration selection and relevance is needed.

-       Specifics on bacterial suspension and OA solution volume are necessary.

-       Duration of staining, washing, and fixation steps should be included.

-       Information on controls and replicates used is essential for result assessment.

-       Overall, these sections need additional details and clarification to ensure consistency, reproducibility, and transparency in the experimental methods and data analysis.

2.6 MTT Assay:

-       The procedure for the MTT assay is generally clear, but it lacks details on certain critical steps, such as the volume of OA solution added to the bacterial suspension and the incubation conditions (e.g., temperature, duration) after adding the MTT solution. Providing this information would ensure consistency and reproducibility of the assay.

-       The section does not mention how the absorbance values obtained from the MTT assay are analyzed or interpreted to assess metabolic activity. Including details on data analysis methods and any statistical analyses performed would enhance the rigor of the study.

3. Results

Overall, the results presented in the manuscript demonstrate the effects of oleanolic acid (OA) on the formation and metabolic activity of Escherichia coli Nissle 1917 (EcN) biofilms, as well as its impact on the growth of EcN and other pathogens. While the results provide valuable insights into the potential applications of OA in promoting probiotic biofilms and inhibiting pathogenic biofilms, there are areas where clarity and detail could be improved:

3.1 Determination of EcN Biofilm Formation Curve:

The results indicate that EcN biomass formation gradually increases with extended cultivation time and stabilizes after 24 hours. However, the data presented in Figure 1b are not described in detail. Providing specific biomass measurements or optical density values at different time points would enhance the clarity and interpretation of the results.

3.2 Determination of Minimum Inhibitory Concentration (MIC) of OA:

The MIC of OA against EcN is reported as 128 μg/mL, and it is mentioned that OA concentrations up to 64 μg/mL do not affect EcN growth. However, it would be beneficial to provide additional details on how the MIC was determined and whether any growth inhibition was observed at concentrations above 128 μg/mL.

3.3 Effects of OA on EcN Biofilm Formation and Pathogen Biofilms:

The results suggest that OA promotes EcN biofilm formation in a concentration-dependent manner, with higher concentrations of OA showing a more pronounced effect. However, the actual biofilm growth percentages provided (e.g., 138.22%, 142.66%) seem unusually high and may require further clarification or validation.

While the inhibitory effects of OA on the biofilm formation of Salmonella 011 and Staphylococcus aureus 002 are described, it would be helpful to include additional data or analysis to support these findings. Providing quantitative measurements of biofilm inhibition and statistical analysis results would strengthen the conclusions.

3.4 Effect of OA on the Metabolic Activity of EcN Biofilm Bacteria:

The results suggest that OA intervention increases the metabolic activity of EcN biofilm bacteria at high concentrations but decreases it at lower concentrations. However, the actual percentage changes in metabolic activity (e.g., 152.83%, 72.81%) seem quite large and may require further validation or explanation.

In summary, while the results provide valuable insights into the effects of OA on EcN biofilms and other pathogens, additional detail, clarification, and validation may be needed to strengthen the conclusions and ensure the reliability of the findings.

4_Discussion

The discussion covers various aspects, including the role of probiotics, the impact of OA on EcN biofilms, the potential mechanisms involved, and future research directions. While comprehensive, organizing the discussion into subsections could improve clarity and readability, making it easier for readers to follow the flow of ideas.

Conciseness: Some sections of the discussion contain repetitive information or details that could be more succinctly summarized. Streamlining the text while retaining essential points would enhance the overall effectiveness of the discussion.

Specific Comments:

-       While the discussion effectively emphasizes the significance of biofilms in probiotic colonization and EcN biofilm promotion, it lacks contextualization regarding challenges in probiotic colonization and biofilm's role in addressing them. Strengthening this aspect would enhance the discussion.

-       The analysis of OA's impact on EcN biofilm formation and inhibition of pathogenic biofilms is comprehensive. However, discussing the concentration-dependent effects of OA in more detail, particularly regarding practical applications, would be beneficial.

-       The discussion outlines potential mechanisms of OA's effects on EcN biofilms but lacks additional context or references to support these hypotheses. Adding supportive evidence would bolster the discussion's robustness.

-       The identification of future research areas is appropriate, but the discussion could elaborate on specific methodologies or experimental approaches to address these gaps, offering more practical guidance.

While the conclusion effectively summarizes the study's main findings and implications, reinforcing key insights derived f

Comments on the Quality of English Language

Moderate editing of English language required

Reviewer 3 Report

Comments and Suggestions for Authors

REVIEW

Dear authors,

The work proposes the use of oleanolic acid as a strong inducer in the formation of biofilm in the commercial probiotic Escherichia coli Nissle 1917 (EcN), which is novel in probiotics since in general the study of biofilm has focused on pathogenic bacteria, being one of the main virulence factors. The in vitro study provides evidence at the transcriptional as well as phenotypic level that oleanolic acid promotes biofilm formation, which benefits EcN for the establishment and potential colonization in the gastrointestinal tract and thus, provide benefits to the host. However, some questions arise about the work, beyond the perspectives they raise.

Please amend the requested comments and submit the revision file.

1.    Would the results obtained from the direct effect of oleanolic acid on EcN be the same if the surface were biotic, that is, if they had used an intestinal epithelial cell line (Caco-2, HT-29)?

2.    I consider that the most appropriate model to compare the biofilm inhibition effect would have been a pathotype of Escherichia coli (EPEC, EHEC), to be able to establish whether the genes involved in bacterial motility are switched off in the same genus and consequently, a change in phenotype.

3.    Would you use oleanolic acid exclusively as an adjuvant for EcN or do you consider it as a potential adjuvant in treatment with any bacterial probiotic?

4.    Among the changes at the transcriptional level evaluated after contact with oleanolic acid, is there a stress on EcN for the production of any bacteriocin?

5.    As mentioned in the discussion, studies should be carried out in adequate in vivo models to evaluate not only the establishment and formation of the probiotic biofilm in the intestine, but also how the administration of oleanolic acid can affect the intestinal microbiota (commensals and pathobionts), as well as metabolic and immunological aspects in situ and systemic in the interaction with the host.

6.    Write in italics the name of the microorganisms and the term in vivo, write CFU in capital letters, homogenize the abbreviation qRT-PCR or RT-qPCR, reduce the font size in the captions of Figure 2 and 5, the term “Pseudotuberculosis” is not written in italics.

Please amend the requested comments and submit the revision file.

Round 2

Reviewer 2 Report

Comments and Suggestions for Authors

Dear Authors,

The manuscript “Oleanolic acid promotes the formation of probiotic Escherichia coli Nissle 1917 (EcN) biofilm by inhibiting bacterial motility” investigates the effect of oleanolic acid (OA) on the formation of biofilm by probiotic Escherichia coli Nissle 1917 (EcN) and explores the underlying mechanisms. 

The last revised version that the authors have submitted is significantly improved, and in my opinion, it can be accepted in its present form

Comments on the Quality of English Language

Moderate editing of English language required

Reviewer 3 Report

Comments and Suggestions for Authors

The observations and suggested corrections were satisfactorily addressed; I consider that the quality of the work is sufficient to be published.